# Short-term health system responses to epidemics across hard to reach areas in sub-Saharan Africa: A scoping review protocol

**Annette A. Murunga** *, **Ben Ngoye**◎, **Francis P. Wafula**◎, **Gilbert O. Kokwaro**◎

Institute of Healthcare Management, Strathmore University Business School, Nairobi, Kenya

◎ These authors contributed equally to this work.
* amurunga@strathmore.edu

## Abstract

Epidemics and Pandemics (disease outbreaks) are the occurrence of cases of disease in excess of what would be normally expected. Epidemic-prone diseases, including emerging and re-emerging diseases, constitute the greatest threat to public health security and disruption of social and economic development. When outbreaks are diagnosed in specific areas, an outbreak response is triggered to stop the spread rapidly. In the past 20 years, the sub-Saharan region has witnessed a marked increase in the number of outbreaks in pandemics, such as cholera, dengue, A/H5N 1 influenza among others. While efforts toward containment have been individually studied, we have no recent studies that examine them collectively in order to draw appropriate comparisons, no recent studies that have especially focused on hard-to-reach areas, and none that have applied a health systems lens. This study thus details a scoping review of short-term health system responses to epidemics across hard-to-reach areas in sub-Saharan Africa. The scoping review will be undertaken following PRISMA guidelines. A modified Donabedian framework will be used to understand the different approaches used while responding to epidemics. The review will focus on published and unpublished studies that report short-term health systems responses to epidemics in hard to reach areas. These will be gleaned from PubMed, google scholar and Cochrane, supplemented by a Google advanced search. In addition, manual searches will be carried out through related articles and websites. Data will be charted, coded, and narratively synthesized. our exclusion criteria will include; protocols, book chapters and countries not identified as hard to reach areas in SSA. We anticipate developing a document that will show the different approaches health systems in different countries used when responding to epidemics. The information generated will contribute to strengthening future epidemic responses by identifying best practices and innovative ideas as well as highlighting knowledge gaps.

## Introduction

The COVID-19 pandemic in 2019 led the world to experience a significant global public health threat that pushed the world to the brink of a social and economic meltdown. This threat

**Data Availability Statement:** No datasets were generated or analysed during the current study. All

relevant data from this study will be made available upon study completion.

**Funding:** 1. A. M- Wellcome (UKAid), Economic and Social Research Council, Medical Research Council- Grant Number MR/T022078/1 2. B.N- Wellcome (UKAid), Economic and Social Research Council, Medical Research Council- Grant Number MR/T022078/1 3. F.W- Wellcome (UKAid), Economic and Social Research Council, Medical Research Council- Grant Number MR/T022078/1 4. G.K- Wellcome (UKAid), Economic and Social Research Council, Medical Research Council- Grant Number MR/T022078/1 https://www.ukri.org/councils/mrc/.

**Competing interests:** The authors have declared that no competing interests exist.

triggered questions about how well the global public health community is prepared to handle major epidemics and pandemics. The COVID-19 pandemic forced countries to adopt drastic response measures, which included imposing total lockdowns due to its rapid person-to-person transmission rates. This was further exacerbated by weak and poorly resourced health systems with inequities especially across low- and middle-income settings most of which are in sub-Saharan Africa put the whole world at risk. Most of these countries also possess pockets of hard-to-reach areas that are described as places where it would be hardest or take the longest for someone to access basic humanitarian services (such as health clinics and hospitals). This access is further compounded by a lack of functioning transport links and infrastructure, as well as terrain difficulty [1].

To contextualize the epidemic challenge better, it is important to realize that health systems in Low-and Middle-Income Countries (LMICs) are barely able to handle health systems challenges in high-access areas such as urban and more affluent areas. Major gaps exist in key areas like surveillance, emergency preparedness and response, risk communication and effective control and management of points of entry [2]. A WHO evaluation reported that most Sub-Saharan Africa (SSA) countries are not equipped to respond to sudden shocks to health systems, yet the region often suffers isolated disease outbreaks [3]. Worse still, broader dynamics continue to contribute to the emergence and re-emergence of outbreaks of new diseases and antimicrobial resistance [4]. Despite the fact that outbreaks such as Severe Acute Respiratory System (SARS), cholera, chikungunya, dengue fever, Rift Valley Fever and Ebola provided warning signs and tested countries' ability to respond [5], COVID-19 still found most LMICs unprepared. And the poor emergency response strategy was made worse by inadequate laboratory and epidemiological surveillance systems and economic, social and political disincentives to case reporting [6].

While COVID-19 presented a particularly unique challenge for most countries, it is pertinent to revert to understanding how SSA countries have been responding to other types of outbreaks. We are particularly keen to document immediate (short-term) responses in order to better understand policy agility and capacity to learn, adapt and change when the need does not seem/appear to be urgent. We know that because most SSA countries focus on urgent rather than important matters, their ability to handle COVID-19, which is both an urgent and important matter, was compromised. We argue that by understanding immediate policy response to outbreaks in low-priority areas, we may be able to characterize and deal with the pain points that prevent countries from prioritizing and establishing mechanisms to nip outbreaks in the bud before the major calamity. We also argue that while effort has gone towards examining health sector responses to specific outbreaks, not enough attention has gone towards characterizing commonalities in immediate response to the disease of different kinds and how the immediate actions translate into (or fail to translate into) medium- and longer-term strategies. Examining the causal relationships between the response activities and the broader health system change is crucial if the global health community is to find a lasting solution to disease outbreaks. Moreover, successful strategies for pandemic response and preparedness over the years have been viewed from a vertical lens that is disease-specific, yet it is plausible that if the response was looked at from a health system lens, the response would be quick and more effective. This explicit linkage appears under-addressed by existing literature.

Our scoping review, therefore, aims to provide an overview of existing evidence on immediate health system responses to epidemics in hard to reach areas in SSA and attempt to answer: what is the context of the health system (Service Delivery, Health workforce, Leadership and Governance, Health financing, Medical products and technology, and Health Information Systems) with regards to pandemic preparedness and management in SSA? What is the diversity of approaches in terms of policies, protocols, communication, community

participation, and surveillance that were used to manage disease outbreaks? And what were the outcomes of the various approaches and models in the early detection of diseases, reduced disease impact on the population, change in health behavior, and strengthened health system?

## The rationale for the scoping review

A scoping review utilizes a systematic and iterative approach to identify and synthesize existing literature or emerging literature on a given topic. For this study, a scoping review will be used to understand the scope of the existing literature on the topic, identify the concepts and definitions used, as well as identify gaps in literature for future areas of scholarship. The scoping review will also be used to examine emerging evidence on short term responses to disease outbreaks.

**Limitations of a scoping review.** Limitations in the use of scoping reviews includes, first developing search terms for a comprehensive search strategy may be difficult since the area of discussion is an emerging area with less well-known information, secondly some of the topical areas or terms such as hard to reach are ill defined and can lead to different definitions for the same topic, thirdly the terms used may not be indexed as Medical Subject Headings may be difficult to find in published literature.

## Conceptual framework for the scoping review

A conceptual framework has been developed using the Donabedian framework [7] of structure, processes and outcomes as illustrated in (Fig 1). The framework is commonly used to assess the quality of health services. For this review, a modified framework will be used to help in understanding approaches used in the immediate health system responses to disease outbreaks. For the review, *Structures* will describe the context of the health system and includes the WHO building blocks of Service Delivery, Human Resource for health (HRH), Leadership Management and Governance (LMG), Health financing, Medical products and technology, Health Information Systems (HMIS) [8]. *The process* will include the functions and the surveillance, policies how to conduct contact tracing, community engagement, and vaccination

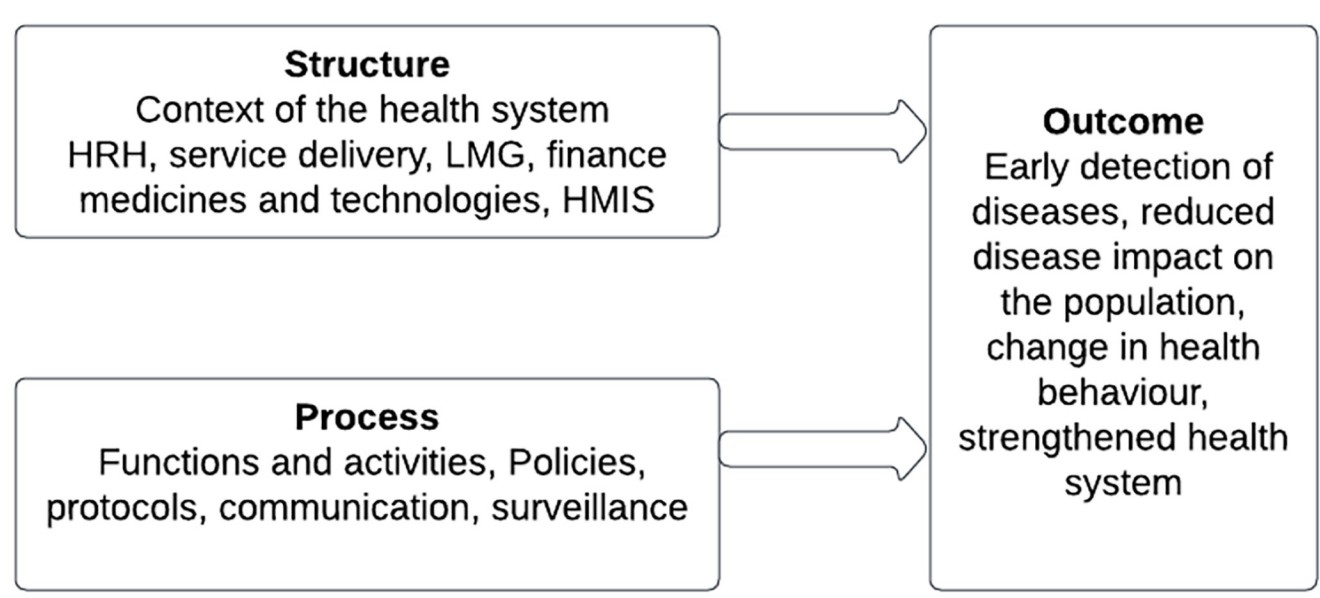

**Fig 1. Modified Donabedian framework for the scoping review.**

campaigns, protocols on health messaging and data collection processes, and communication between partners, and *outcomes* will include surveillance and early detection of disease outbreaks, supply chain management, treatment centers, community engagement, contact tracing, rapid response teams, public health messaging, data collection and analysis, Vaccination campaigns (if applicable), coordination between institutions and different organizations (national, International, NGOs).

Aided by this modified framework, we focus on the following review questions:

1. Context of the health system building block: What contextual forces are shaping the health system, and what are the key challenges faced by countries hard to reach areas in SSA?

2. Processes (functions and activities) of the health system: What are the main functions and activities of the health system hard to reach areas in SSA, and what diversity of approaches, models, innovations and tools were used to manage epidemics?

3. Outcomes of the health system: How successful are the various approaches and models in improving preventing and managing disease epidemics hard to reach areas in SSA?

## Methodology

The review seeks to describe the short-term health system responses to epidemics across hard-to-reach areas in sub-Saharan Africa. Like other scoping reviews, the purpose is to broadly describe a phenomenon (what is known and what isn't) and characterize it in ways that contribute towards a more detailed understanding and/or further research. Consequently, we seek to identify and map available evidence from a wide variety of sources, including published and unpublished material. We follow Arksey and O'Malley's five-stage scoping review framework [9], which proposes the following five stages: identifying the research question, finding the articles, selecting appropriate articles, charting the data, and collating and summarizing the data into meaningful themes.

Stage 1: Identifying the research question

The research question in this review is based on the Joanna Briggs Institute PCC (Population-concept-context) model [10]. An overarching research question was developed to guide the search strategy: "What are the short-term health system responses to epidemics across *hard to reach areas* in sub-Saharan Africa?". The question allows us to capture appropriate existing literature and also provides an opportunity for research questions to be added or modified in the course of the study.

Stage 2: Identifying relevant studies

Our approach will include a systematic search of peer-reviewed studies from electronic databases—PubMed, Google Scholar and Cochrane, snowballing from article reference lists to include additional studies that may not have been indexed. We will also search gray literature using Google. Literature search strategies will be developed using medical subject headings (MeSH-terms) and text words related to the population, geographical terms where countries identified as *hard to reach areas* will be analyzed and phenomenon of interest/intervention, as shown in Table 1.

Stage 3: Selecting articles.

A three-step search strategy will be undertaken. An initial search of three databases will be used that is PubMed, google scholar and Cochrane gray literature will also be accessed through google, this will be followed by an analysis of the text words contained in the title and abstract and the index terms used. The second step will include using all identified keywords and index

**Table 1. Search terms used in searching Pub-Med electronic database.**

| Group A: Target population terms (combined by 'OR') | Group B: Geographical terms (combined by 'OR') | Group C: intervention (combined by 'OR') |
|---|---|---|
| AND<br>cholera OR diarrhoea OR diarrhoea OR dysentery OR Covid-19 OR leishmaniasis OR kala azar OR trypanosomiasis OR sleeping sickness OR malaria OR polio OR measles OR mumps OR meningitis OR chikungunya OR yellow fever OR dengue OR Ebola OR SARS OR MERS OR influenza OR Zika OR epidemic | AND<br>Angola OR Benin OR Burkina Faso OR Burundi OR Central African Republic OR Chad OR Comoros OR the Democratic Republic of Congo OR Djibouti OR Equatorial Guinea OR Eritrea OR Ethiopia OR Gambia OR Guinea OR Guinea-Bissau OR Lesotho OR Liberia OR Madagascar OR Malawi OR Mali OR Mozambique OR Niger OR Sao Tome and Principe OR Senegal OR Sierra Leone OR Somalia OR Sudan OR Tanzania OR Togo OR Uganda OR Zambia OR OR Rwanda OR Congo OR Côte d'Ivoire OR Ivory Coast OR Ghana OR Zimbabwe OR Namibia OR Swaziland OR Botswana OR Gabon OR Mauritius OR Kenya OR Sub-Saharan Africa OR less developed countries in Sub-Saharan Africa | Health systems impact OR Health systems resilience OR Health systems performance OR Health systems response OR Health system assessment, OR Immediate health system response, OR Short-term health system response |

terms in all the included databases and gray literature. The third step will include a manual review of the reference list of all the studies for additional articles that may have been left out during the initial search. This will be done firstly by searching through the reference lists of all included articles, secondly citation tracking in which we shall track selected articles that cite each one of the included articles, and thirdly similar to the citation tracking, we will follow all "related to" or "similar" articles where data collected does not have the word hard to reach areas, data will be collected based on countries included in the study.

## Inclusion criteria

The inclusion criteria will include empirical studies with either qualitative or quantitative data published in English as well as countries termed as less developed countries by the United Nations. Systematic and scoping reviews will also be considered. Full text English papers from between 2010 to 2024 will be considered. This period is critical because majority of the major disease outbreaks occurred during this period.

## Exclusion criteria

The scoping review will exclude all types of reviews, protocols, book chapters and countries not identified as *hard to reach areas* in SSA

MeSH terms used to search for CAB Direct included: *Epidemic OR disease outbreak, AND Health system response AND hard to reach Countries AND Sub-Saharan Africa*; and for Google Scholar, *Health system response to epidemics AND hard to reach AND Sub-Saharan Africa*

Following the search strategy above, all identified articles will be uploaded onto Zotero, where all duplicates will be removed.

The first screening process will include reading the title and the abstracts to reach the following decisions: (1) if at least one reviewer agrees to include or consider the abstract or title to be inconclusive, the study will be moved to the next level of screening; (2) for any of the studies if both reviewers agree for an article to be excluded, the article will be excluded. All studies at this level will be charted in an Excel sheet.

In the second level of screening, the full texts will be assessed by two independent reviewers. Reasons for any exclusion will be recorded and reported in the scoping review. Any disagreements will be resolved through discussions or with a third reviewer. The results of the search

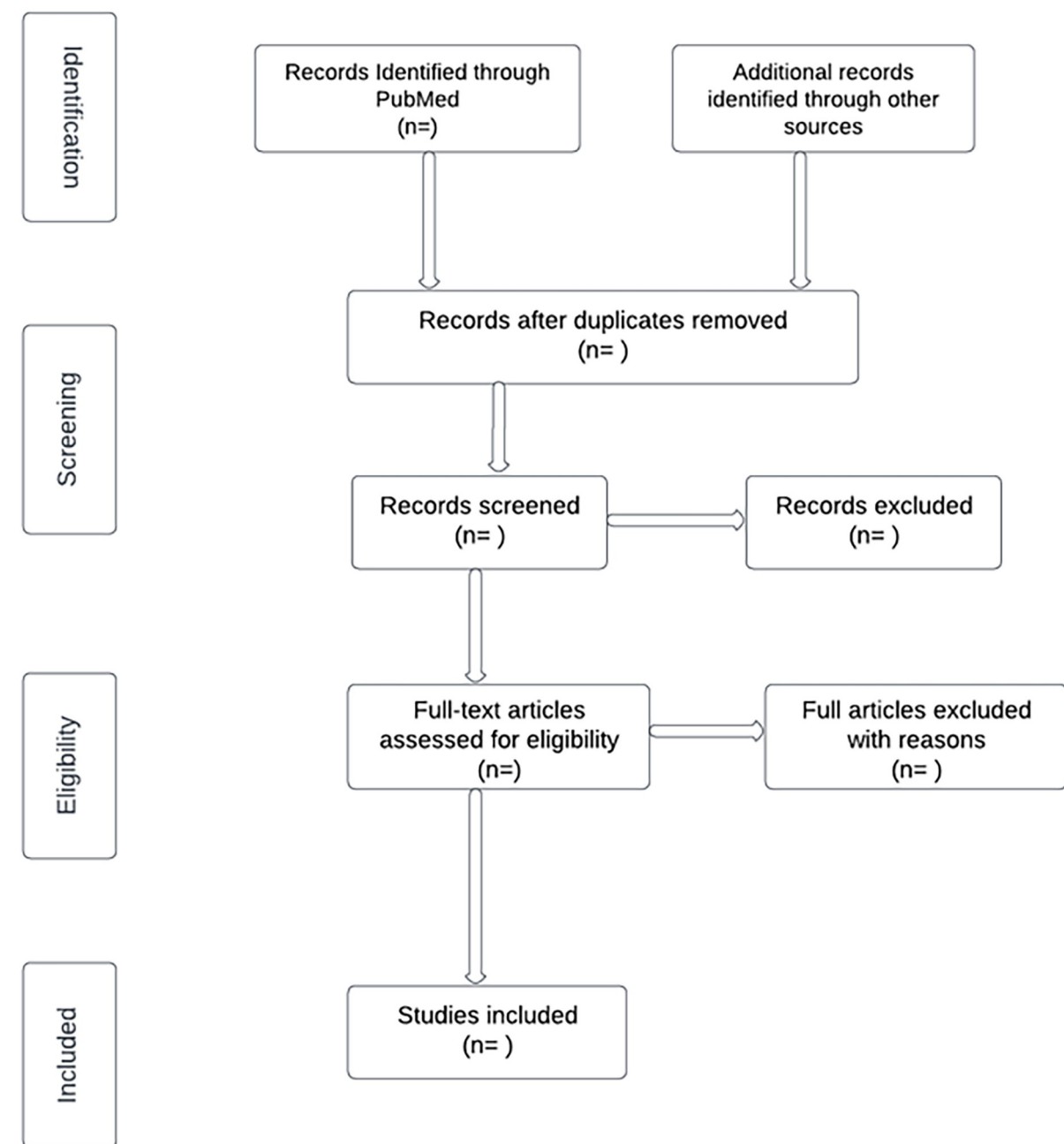

**Fig 2. Preferred Reporting Items for Systematic Reviews and Meta-Analyses (PRISMA) flow diagram.**

and study selection will be reported in a Preferred Reporting Items for Systematic Reviews and Meta-Analyses (PRISMA) as illustrated in the flow diagram (Fig 2).

Stage 4: Charting data

To answer the research question, a data charting form in Excel with the following features will be created: authors, year of publication, study title, study design, the aim of the study, country of the study, and key findings relevant to the review objective. A draft charting table is provided (S1 Appendix). As a preliminary step, the reviewers will independently extract data from the first five articles using the data-charting table and meet to determine whether the

approach to data extraction is consistent with the purpose of the study. The draft data extraction tool will be modified and revised as necessary during the extraction process. Modifications will be detailed in the full scoping review article.

Stage 5: collating, summarizing and reporting the results

The data from stage 4 will be collated, summarized and reported in a manner that aligns with the study objectives. Tabular and graphical representations of the data may be used to illustrate the identified results and will be supported by narrative descriptions of the data. The meaning of the findings and implications for future research and practice will be discussed.

## Patient and public involvement

Patients and the public will not be involved in the design and conception of the study.

## Ethics and dissemination

The proposed scoping review does not require ethical approval as data will be collected through published peer-reviewed literature and grey literature. This scoping review will provide a comprehensive overview of existing evidence in the field and will be disseminated to relevant public health offices, during conferences and stake holder meetings.

## Supporting information

**S1 Appendix.**
(DOCX)

## Author Contributions

**Conceptualization:** Annette A. Murunga, Francis P. Wafula.

**Data curation:** Annette A. Murunga.

**Formal analysis:** Annette A. Murunga.

**Funding acquisition:** Annette A. Murunga, Ben Ngoye, Francis P. Wafula, Gilbert O. Kokwaro.

**Investigation:** Annette A. Murunga, Ben Ngoye.

**Methodology:** Annette A. Murunga.

**Project administration:** Annette A. Murunga, Ben Ngoye, Gilbert O. Kokwaro.

**Supervision:** Ben Ngoye, Francis P. Wafula, Gilbert O. Kokwaro.

**Validation:** Ben Ngoye.

**Writing – original draft:** Annette A. Murunga.

**Writing – review & editing:** Ben Ngoye, Francis P. Wafula, Gilbert O. Kokwaro.

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
