## [Decision Letter · Decision Letter 0]

13 Sep 2023

PONE-D-23-13244Short-term health system responses to epidemics across hard-to-reach areas in sub-Saharan Africa: A scoping review protocolPLOS ONE

Dear Ms. Murunga,

Thank you for submitting your manuscript to PLOS ONE. After careful consideration, we feel that it has merit but does not fully meet PLOS ONE’s publication criteria as it currently stands. Therefore, we invite you to submit a revised version of the manuscript that addresses the points raised during the review process.

Thanks for submitting your protocol. After careful review, I recommend you address the comments provided by the reviewer.

In addition, please make sure that you address the following questions:

Review and discuss your choice to search only a few databases while leaving out other important sources for biomedical literature and research.Discuss any anticipated limitations to the method selected for your scoping review. Please include a section on the limitations and discuss any challenges that could limit a comprehensive retrieval of the literature (e.g., the review will consider manuscripts published in English only, only a few databases to be used for manuscript search, etc.) ==============================

We look forward to receiving your revised manuscript.

Kind regards,

Anselme Shyaka, Ph.D

Academic Editor

PLOS ONE

Journal Requirements:

2. Please include a caption for figure 1.

Reviewers' comments:

Reviewer's Responses to Questions

**Comments to the Author**

1. Does the manuscript provide a valid rationale for the proposed study, with clearly identified and justified research questions?

Reviewer #1: Yes

2. Is the protocol technically sound and planned in a manner that will lead to a meaningful outcome and allow testing the stated hypotheses?

Reviewer #1: Yes

3. Is the methodology feasible and described in sufficient detail to allow the work to be replicable?

Reviewer #1: Yes

4. Have the authors described where all data underlying the findings will be made available when the study is complete?

Reviewer #1: Yes

5. Is the manuscript presented in an intelligible fashion and written in standard English?

Reviewer #1: Yes

6. Review Comments to the Author

You may also provide optional suggestions and comments to authors that they might find helpful in planning their study.

Reviewer #1: Overall, this scoping review protocol appears well-structured and comprehensive, and it provides a clear outline of how the study will be conducted.

However, here some proposed areas for improvement:

Major comment:

P.7 - There should be an alignment between the title – main objective - and the methodology. The methodology outlines a clear and well-structured approach for conducting a scoping review and provides details on the research questions, search strategy, data selection, and data extraction. However, it is unclear how the search approach proposed will collect data on hard-to-reach areas specifically. Details on specific search for these areas should be defined. This is important since these areas may not necessarily be characterized as ‘hard - to- reach' in literature.

Minor comments

Abstract & P.3 - The use of the word ‘outbreak’ should be reviewed. This is a general term that englobes both ‘epidemic’ and ‘pandemic’. Ensure you use it appropriately and replace it epidemics and/or pandemics whenever applicable.

Abstract - There is contradiction in this sentence. If the objective is to stop infection, the target is MANY people, not few. But this sentence can be better formulated. Suggestion: use the term 'minimize infections'.

Abstract - Avoid starting sentences with 'And'.

Abstract - This abstract should briefly mention the methodology.

P.3 - Provide more details and clarification on how the manual search will be carried out.

P. 4 - The use of 'short-term' two times in one sentence is redundant. Reformulate.

P.5 - Capitalize ‘COVID-19’

P.5 - Explain the full meaning of abbreviations the first time they are used (eg: SSA, LMICs, SARS, etc). They can simply be used as abbreviations afterwards.

P.6 - Reformulate this sentence ‘To contextualize the challenge better, though, it is important to realize that health systems in LMICs are barely able to handle those in high-access areas such as urban and more affluent Areas.’ It is unclear what ‘those’ is replacing.

P.6 - 7. The rationale for scoping review as it is presented gives a broad definition of scoping review and a difference between scoping review and systematic review. Define the rationale specifically for this specific scoping review; the needs and expected outcomes.

P.7- This part - as for others - should be aligned with the short-term aspect of your research project. Some of the outcomes mentioned here are long-term. Short-term health system responses include:

-Surveillance and early detection

-Rapid response teams

-Treatment Centers

- Public Health Messaging

- Supplying Chain Management

- Community Engagement

- Contact Tracing

- Data Collection and Analysis

- Vaccination campaigns (if applicable)

- Coordination between institutions and different organizations (national, International, NGOs).

- Explore ways of incorporating these in the Process.

P. 8- It is unclear what will be done in this step of screening the reference list. Provide more details.

P.10 - Define inclusion and exclusion criteria.

7. PLOS authors have the option to publish the peer review history of their article (what does this mean?). If published, this will include your full peer review and any attached files.

Reviewer #1: No

---

## [Author Response · Author response to Decision Letter 0]

8 Feb 2024

1. Review and discuss your choice to search only a few databases while leaving out other important sources for biomedical literature and research.

Ans: The study focused on three databases that is PubMed, Google Scholar and Cochrane. The databases were selected on the basis that 1) Google Scholar has a wide variety of articles and is also very comprehensive that is it includes literature that goes beyond biomedical research. The data base also provides provision for gray literature. PubMed on the other hand has a free search interface and provides access to more than 26million citations covering biomedical literature. The literature provided dates back from 1946 in MEDLINE AND PreMEDLINE. The database also includes publisher-supplied records (including electronic publications ahead of print) (1).

2. Discuss any anticipated limitations to the method selected for your scoping review. Please include a section on the limitations and discuss any challenges that could limit a comprehensive retrieval of the literature (e.g., the review will consider manuscripts published in English only, only a few databases to be used for manuscript search, etc.)

Limitations in the use of scoping reviews includes, first developing search terms for a comprehensive search strategy may be difficult since the area of discussion is an emerging area with less well-known information, secondly some of the topical areas or terms such as hard to reach are ill defined and can lead to different definitions for the same topic, thirdly the terms used may not be indexed as Medical Subject Headings may be difficult to find in published literature.

P.7 - There should be an alignment between the title – main objective - and the methodology. The methodology outlines a clear and well-structured approach for conducting a scoping review and provides details on the research questions, search strategy, data selection, and data extraction. However, it is unclear how the search approach proposed will collect data on hard-to-reach areas specifically. Details on specific search for these areas should be defined. This is important since these areas may not necessarily be characterized as ‘hard - to- reach' in literature.

Ans: Thank you for taking note of this. The term hard to reach areas has now been defined are as places where it would be hardest or take the longest for someone to access basic humanitarian services (such as health clinics and hospitals). This access is further compounded by a lack of functioning transport links and infrastructure, as well as terrain difficulty. The countries to be included in this category include: 

Angola Ethiopia Rwanda

Benin Gambia São Tomé and Príncipe

Burkina Faso Guinea Senegal

Burundi Guinea-Bissau Sierra Leone

Central African Republic Lesotho Somalia

Chad Liberia Sudan

Comoros Madagascar Tanzania

Democratic Republic of the Congo Malawi Togo

Djibouti Mali Uganda

Equatorial Guinea Mauritania Zambia

Eritrea Mozambique Rwanda

 Niger São Tomé and Príncipe

Abstract & P.3 - The use of the word ‘outbreak’ should be reviewed. This is a general term that englobes both ‘epidemic’ and ‘pandemic’. Ensure you use it appropriately and replace it epidemics and/or pandemics whenever applicable.

Ans Replaced with Epidemic

Abstract - There is contradiction in this sentence. If the objective is to stop infection, the target is MANY people, not few. But this sentence can be better formulated. Suggestion: use the term 'minimize infections'-this has been done.

Abstract - Avoid starting sentences with 'And'-this has been corrected

Abstract - This abstract should briefly mention the methodology-this has been corrected and added

P.3 - Provide more details and clarification on how the manual search will be carried out.

The third step will include a manual review of the reference list of all the studies for additional articles that may have been left out during the initial search. This will be done firstly by searching through the reference lists of all included articles, secondly citation tracking in which we shall track selected articles that cite each one of the included articles, and thirdly similar to the citation tracking, we will follow all “related to” or “similar” articles where data collected does not have the word hard to reach areas, data will be collected based on countries included in the study.

P. 4 - The use of 'short-term' two times in one sentence is redundant. Reformulate this has been reformulated

Our scoping review, therefore, aims to provide an overview of existing evidence on immediate health system responses to epidemics in hard to reach areas in SSA and attempt to answer: what is the context of the health system (Service Delivery, Health workforce, Leadership and Governance, Health financing, Medical products and technology, and Health Information Systems) with regards to pandemic preparedness and management in SSA? What is the diversity of approaches in terms of policies, protocols, communication, community participation, and surveillance that were used to manage disease ou

tbreaks? And what were the outcomes of the various approaches and models in the early detection of diseases.

P.5 - Capitalize ‘COVID-19-Done

P.5 - Explain the full meaning of abbreviations the first time they are used (eg: SSA, LMICs, SARS, etc). They can simply be used as abbreviations afterwards. Done

P.6 - Reformulate this sentence ‘To contextualize the challenge better, though, it is important to realize that health systems in LMICs are barely able to handle those in high-access areas such as urban and more affluent Areas.’ It is unclear what ‘those’ is replacing.

To contextualize the epidemic challenge better, it is important to realize that health systems in Low-and Middle-Income Countries (LMICs) are barely able to handle health systems challenges in high-access areas such as urban and more affluent areas.

P.6 - 7. The rationale for scoping review as it is presented gives a broad definition of scoping review and a difference between scoping review and systematic review. Define the rationale specifically for this specific scoping review; the needs and expected outcomes.

A scoping review utilizes a systematic and iterative approach to identify and synthesize existing literature or emerging literature on a given topic. For this study, a scoping review will be used to understand the scope of the existing literature on the topic, identify the concepts and definitions used, as well as identify gaps in literature for future areas of scholarship.

P.7- This part - as for others - should be aligned with the short-term aspect of your research project. Some of the outcomes mentioned here are long-term. Short-term health system responses include:

-Surveillance and early detection

-Rapid response teams

-Treatment Centers

- Public Health Messaging

- Supplying Chain Management

- Community Engagement

- Contact Tracing

- Data Collection and Analysis

- Vaccination campaigns (if applicable)

- Coordination between institutions and different organizations (national, International, NGOs).

- Explore ways of incorporating these in the Process

This has been done.

p.10-Define the inclusion and exclusion criteria.

Inclusion criteria

The inclusion criteria will include empirical studies with either qualitative or quantitative data published in English as well as countries termed as less developed countries by the United Nations. Systematic and scoping reviews will also be considered. Full text English papers from between 2010 to 2024 will be considered. This period is critical because majority of the major disease outbreaks occurred during this period.

Exclusion criteria

The scoping review will exclude all types of reviews, protocols, book chapters and countries not identified as hard to reach areas in SSA

---

## [Editor Report · Decision Letter 1]

30 Jul 2024

Short-term health system responses to epidemics across hard to reach areas in sub-Saharan Africa: A scoping review protocol

PONE-D-23-13244R1

Dear Authors

We’re pleased to inform you that your manuscript has been judged scientifically suitable for publication and will be formally accepted for publication once it meets all outstanding technical requirements.

Kind regards,

Raquel Inocencio da Luz, Phd

Academic Editor

PLOS ONE
---

## [Editor Report · Acceptance letter]

2 Aug 2024

PONE-D-23-13244R1 

PLOS ONE

Dear Dr. Murunga, 

I'm pleased to inform you that your manuscript has been deemed suitable for publication in PLOS ONE. Congratulations! Your manuscript is now being handed over to our production team.

Kind regards, 

on behalf of

Dr Raquel Inocencio da Luz 

Academic Editor

PLOS ONE